# EcoVLA: Environment-Aware Adaptive Pruning with Interleaved Inference Orchestration for Vision-Language-Action Models

Yuting Huang [* 1]  Leilei Ding [* 1]  Zhipeng Tang [* 1]  Zenghuan Zhu [1]  Jiajun Deng [1]  Xinrui Lin [1]  Shuo Liu [1]  Haojie Ren [1]  Jianmin Ji [1]  Yanyong Zhang [1]

## Abstract

While Vision-Language-Action (VLA) models hold promise in embodied intelligence, their large parameter counts lead to substantial inference latency that hinders real-time manipulation, motivating parameter sparsification. However, as the environment evolves during VLA execution, the optimal sparsity patterns change accordingly. Static pruning lacks the adaptability required for environment dynamics, whereas fixed-interval dynamic layer pruning suffers from coarse granularity and high retraining overheads. To bridge this gap, we propose **EcoVLA**, a training-free, plug-and-play adaptive pruning framework that supports orthogonal combination with existing VLA acceleration methods. EcoVLA comprises two components: **E**nvironment-aware **A**daptive **P**runing (**EAP**) and **I**nterleaved **I**nference **O**rchestration (**I²O**). EAP is a lightweight adaptive channel pruning method that incorporates the temporal consistency of the physical environment to update sparsity patterns. I²O leverages the FLOPs bubbles inherent in VLA inference to schedule the pruning method in parallel, ensuring negligible impact on latency. Evaluated on diverse VLA models and benchmarks, EcoVLA delivers state-of-the-art performance, achieving up to $1.60\times$ speedup with only a 0.4% drop in success rate, and further reaches $2.18\times$ speedup with only a 0.5% degradation when combined with token pruning. We further validate the effectiveness of EcoVLA on real-world robots. Our code is available here.

## 1. Introduction

Vision-Language-Action (VLA) models enhance the generalization of embodied intelligence by injecting semantic understanding into robot control (Black et al., 2024; Li et al., 2024a; Kim et al., 2024). Despite promising real-world results from recent VLA models such as OpenVLA (Kim et al., 2024) and $\pi_{0.5}$ (Intelligence et al., 2025), inference latency remains the primary bottleneck for real-time control (Yang et al., 2025). In this context, token pruning (Wang et al., 2025; Liu et al., 2025) has been extensively researched as a method to reduce input size, which decreases latency by discarding redundant visual tokens at the risk of losing critical semantics with high pruning ratio. While token-level optimization has been thoroughly studied (Shinde et al., 2025), comparatively little attention has been paid to VLA model pruning. In this work, we focus on VLA model pruning, an area with less research, to further accelerate inference by reducing redundant model parameters. Typically, a VLA architecture comprises a lightweight action expert and a heavy VLM backbone, which dominates the parameter count and shows high sparsity, making it a prime target for pruning (Chen & Li, 2025; Jabbour et al., 2025).

Existing research on VLA model sparsification primarily bifurcates into two directions, static and dynamic pruning. Static pruning methods, such as RLRC (Chen & Li, 2025) and GLUESTICK (Jabbour et al., 2025), perform pruning or recovery offline. However, they fail to adapt to the dynamically evolving task environment (e.g., transitioning from large-scale navigation to fine-grained local manipulation), where optimal sparsity patterns vary dynamically (Liu et al., 2023b), as shown in Figure 1. Consequently, these static approaches often suffer performance degradation under dynamic sparsity shifts. Moreover, they are heavily shackled by the requirement for massive retraining cycles or a reconstruction process that incurs unsustainable computational costs. To address the inflexibility of static approaches, dynamic pruning methods like MoLe-VLA (Zhang et al., 2026) and DeeR-VLA (Yue et al., 2024) select layers at fixed intervals based on runtime inputs but suffer from critical drawbacks: the dependency on auxiliary routers incurs extra training and runtime inference overheads, while their

*Equal contribution  [1]University of Science and Technology of China, Hefei, China. Correspondence to: Jianmin Ji <jianmin@ustc.edu.cn>, Yanyong Zhang <yanyongz@ustc.edu.cn>.

*Proceedings of the 43rd International Conference on Machine Learning*, Seoul, South Korea. PMLR 306, 2026. Copyright 2026 by the author(s).

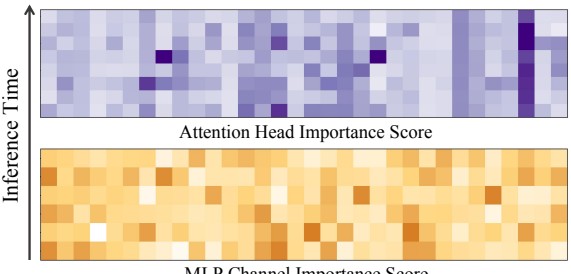

*Figure 1.* During VLA execution, channel importance scores vary dynamically as the environment evolves, causing the optimal sparsity pattern to shift accordingly.

coarse layer-level granularity overlooks fine-grained intra-layer redundancy.

To bridge these gaps, a training-free, fine-grained, and environment-aware adaptive model pruning method is urgently needed. However, two major challenges remain. First, VLA models' sparsity patterns evolve with the environment, making real-time computation difficult. Additionally, relying only on instantaneous observations fails to capture the continuous nature of VLA execution. Second, adaptive pruning introduces real-time overhead. While existing methods for LLMs use large-batch inference to amortize overhead across multiple samples (Le et al., 2025), VLA models constrained by single-sample streaming bear the pruning overhead individually, directly adding it to the end-to-end inference latency. For frequency-sensitive VLA models, such delays limit the effective policy update rate, inducing robotic stuttering and jittering (Black et al., 2026).

To address these challenges, we propose **EcoVLA**, a training-free, plug-and-play pruning framework capable of adapting sparsity patterns via real-time environmental perception while minimizing pruning overhead through non-blocking parallel inference. EcoVLA comprises two core components: **E**nvironment-aware **A**daptive **P**runing (**EAP**) and **I**nterleaved **I**nference **O**rchestration (**I²O**).

EAP is a lightweight, environment-aware adaptive structured channel pruning method. First, leveraging visual observations, EAP perceives environmental dynamics to identify variations in sparsity patterns. Crucially, to maintain the temporal consistency essential for stable VLA execution, we incorporate a temporal feature aggregation strategy. By strategically integrating the instantaneous features with historical features, EAP precisely identifies redundant channels for pruning. The continuous update of historical features with the latest features further guarantees the temporal consistency of the sparsity patterns.

I²O replaces the conventional sequential paradigm with a non-blocking parallel paradigm, strategically exploiting FLOPs Bubbles within VLA inference. Specifically, I²O orchestrates two parallel streams, comprising an Inference Stream for real-time action generations and a Pruning Stream for pruning execution. By interleaving pruning computations into the FLOPs bubbles, we maximize overall hardware utilization, effectively masking the pruning overhead to ensure robust real-time streaming control. We evaluate EcoVLA on robotic manipulation across two simulators (LIBERO (Liu et al., 2023a) and SIMPLER (Li et al., 2024b)) and three VLA models (OpenVLA-OFT (Kim et al., 2025), $\pi_{0.5}$ (Intelligence et al., 2025) and CogACT (Li et al., 2024a)). EcoVLA achieves $1.6\times$ speedup with only a $0.4\%$ reduction in success rate. Furthermore, by integrating with token pruning methods, EcoVLA boosts the speedup from $1.21\times$ (achieved by FastV with a $50\%$ pruning ratio) to $2.18\times$. Crucially, it recovers the accuracy drop caused by FastV, narrowing the performance gap with the vanilla baseline to just $0.5\%$. To validate our approach beyond simulation, we deploy EcoVLA on a 7-DoF Kinova Gen3 and a 19-DoF Franka equipped with XHand, demonstrating its practical acceleration capabilities in real-world scenarios.

## 2. Related Work

**Vision-Language-Action Models.** With the rapid development of embodied intelligence, research attention has increasingly expanded beyond high-level task planning toward embodied task execution (Huang et al., 2025; Lin et al., 2025; Kim et al., 2024). In this context, VLAs extend VLMs by incorporating action modalities for embodied control, while also integrating certain planning capabilities to bridge perception, reasoning, and action (Brohan et al., 2022; Black et al., 2024; Kim et al., 2025; Tang et al., 2025b). Despite their efficacy, deployment is constrained by high computational costs (Yang et al., 2025; Jiang et al., 2026). As action heads (e.g., MLPs, diffusion) are lightweight, the VLM backbone remains the dominant computational bottleneck (Zhang et al., 2026; Ma et al., 2025).

**Efficient Vision-Language-Action Models.** VLA acceleration typically employs token pruning (Wang et al., 2025; Liu et al., 2025) or KV caching (Xu et al., 2026) to exploit input redundancy. For model sparsification, static methods (Chen & Li, 2025; Jabbour et al., 2025) prune parameters offline but lack environmental adaptability and require retraining or reconstruction. Conversely, fixed-interval dynamic methods (Yue et al., 2024; Zhang et al., 2026) offer task-dependence but suffer from high retraining costs and limited transferability. While other techniques such as action chunking (Zhao et al., 2023), which predicts multiple steps of actions at once, and asynchronous inference (Tang et al., 2025a; Black et al., 2026), which overlaps model inference with action execution, can also yield speedup benefits, they are not designed to address these limitations. Consequently, a training-free, fine-grained, and plug-and-play adaptive pruning framework is needed, which can be readily integrated with existing acceleration methods.

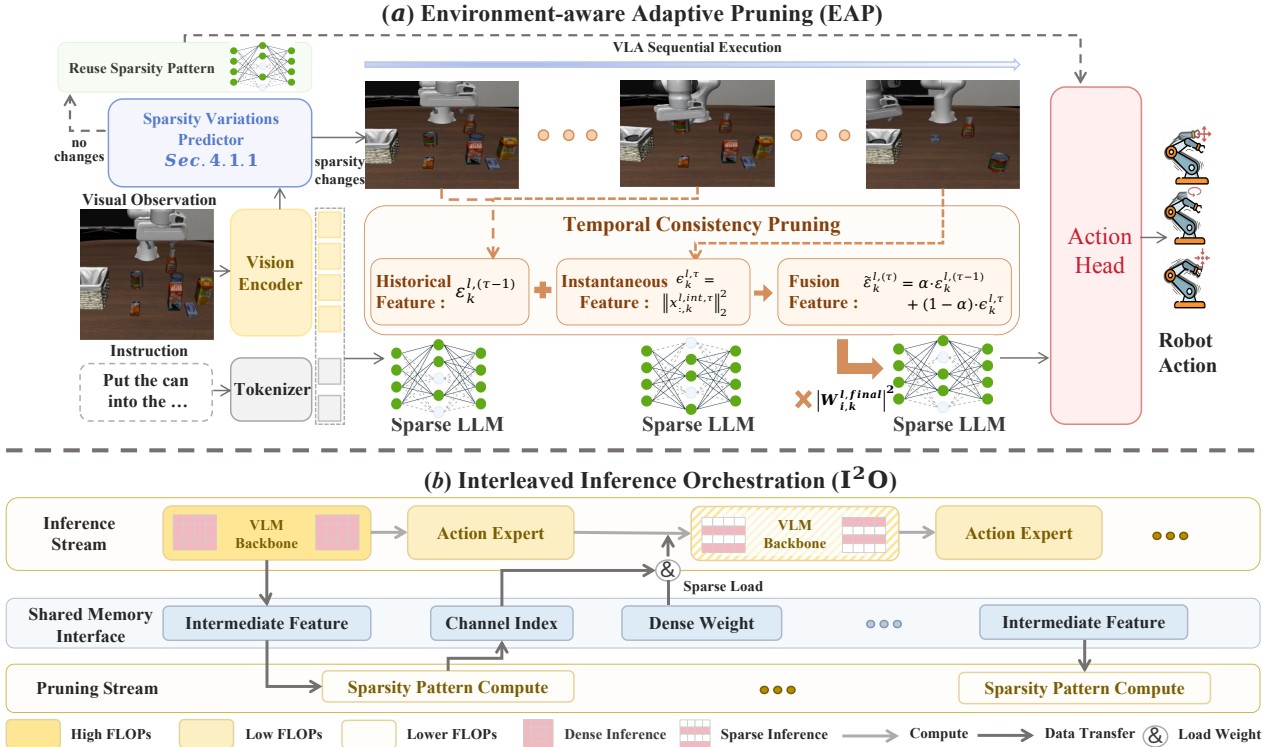

*Figure 2.* Overall pipeline of EcoVLA. (a) **E**nvironment-aware **A**daptive **P**runing (**EAP**): EAP is a lightweight, environment-aware method that identifies sparsity variations by perceiving real-time dynamics. Considering the temporal consistency of VLA execution in physical environments, EAP integrates instantaneous features with historical features to jointly compute the sparsity pattern. (b) **I**nterleaved **I**nference **O**rchestration (**I²O**): I²O interleaves sparsity pattern computation into the inherent FLOPs bubbles within the VLA inference using a non-blocking parallel paradigm.

## 3. Preliminaries

### 3.1. Vision-Language-Action Models

Vision-Language-Action (VLA) models unify visual and linguistic inputs to generate robotic actions, typically consisting of a Vision-Language Model (VLM) backbone and an Action Expert (Kim et al., 2025; Wen et al., 2025). While Action Expert architectures vary (e.g., Diffusion-based or Parallel Decoding) (Black et al., 2024; Kim et al., 2025; Intelligence et al., 2025), they are lightweight compared to the VLM backbone. Thus, the VLM constitutes the primary bottleneck, rendering its standardized architecture the ideal candidate for pruning.

### 3.2. Formulation of VLA Structured Pruning

Let $\pi_\Theta$ be a VLA model with Large Language Model (LLM) backbone $\Theta$. We target $\Theta$ for pruning due to its dominant model size. We optimize a structural binary mask $\mathbf{m} \in \{0, 1\}^{|\Theta|}$ (where grouped elements share values) to minimize the divergence between dense and pruned policies:

$$\arg\min_{\mathbf{m}} \mathcal{L}\left(\pi_\Theta, \pi_{\Theta \odot \mathbf{m}}, \mathcal{D}\right) \quad \text{s.t.} \quad \|\mathbf{m}\|_0 = \kappa \quad (1)$$

where $\odot$ denotes the element-wise product, $\kappa$ is sparsity constraint, and $\mathcal{D}$ represents the calibration dataset. The objective $\mathcal{L}$ measures the divergence between models.

To efficiently solve the global optimization problem defined in (1), we decompose the LLM into layer-wise reconstruction paradigm (An et al., 2024; Le et al., 2025). The LLM consists of $L$ blocks. Each block $l$ transforms the input hidden state $\mathbf{X}^l \in \mathbb{R}^{B \times S \times D}$ via a residual mapping:

$$\mathbf{X}^{l+1} = \mathbf{X}^l + \mathcal{F}^l(\mathbf{X}^l) \quad (2)$$

We decompose the block function $\mathcal{F}^l$ into an intermediate transformation and a final linear projection:

$$\mathcal{F}^l(\mathbf{X}^l) = \mathbf{X}^{l,\text{int}}(\mathbf{W}^{l,\text{final}})^T, \qquad \mathbf{X}^{l,\text{int}} = \mathcal{T}^l(\text{LN}(\mathbf{X}^l)) \quad (3)$$

Here, $\mathcal{T}^l$ is intermediate transformation (e.g., $\mathbf{W}^K$ or $\mathbf{W}^{\text{up}}$), while $\mathbf{W}^{l,\text{final}}$ represents the final weight matrix (e.g., $\mathbf{W}^O$ or $\mathbf{W}^{\text{down}}$). Since these intermediate representations encode rich contextual features (Ding et al., 2024; Yan et al., 2026), they can be effectively utilized to guide the computation of the structured sparsity patterns. For hardware acceleration, structured pruning aligns output channels with input

channels, retaining indices $\mathbb{C}^l \subseteq \{1, 2, \ldots, C_{\text{in}}\}$:

$$\widetilde{\mathbf{W}}^{l,\text{gate}} = \mathbf{W}^{l,\text{gate}}[\mathbb{C}^l, :], \quad \widetilde{\mathbf{W}}^{l,\text{up}} = \mathbf{W}^{l,\text{up}}[\mathbb{C}^l, :],$$
$$\widetilde{\mathbf{W}}^{l,\text{down}} = \mathbf{W}^{l,\text{down}}[:, \mathbb{C}^l], \tag{4}$$

where $\widetilde{\mathbf{W}}^{l,\text{gate}}, \widetilde{\mathbf{W}}^{l,\text{up}} \in \mathbb{R}^{|\mathbb{C}^l| \times C_{\text{out}}}$ and $\widetilde{\mathbf{W}}^{l,\text{down}} \in \mathbb{R}^{C_{\text{out}} \times |\mathbb{C}^l|}$. The notation $|\mathbb{C}^l|$ is the cardinality of $\mathbb{C}^l$. Similarly, in attention blocks, pruning heads removes coupled output channels of $\mathbf{W}^{Q,K,V}$ and input channels of $\mathbf{W}^O$.

# 4. Methodology

In this section, we introduce EcoVLA, the first training-free, plug-and-play adaptive pruning framework for VLA models, as illustrated in Figure 2. We first detail Environment-aware Adaptive Pruning (EAP) in Section 4.1, which begins with a lightweight predictor to identify real-time sparsity variations and subsequently employs a pruning method based on temporal consistency. Next, we introduce Interleaved Inference Orchestration (I²O) in Section 4.2, a parallel execution paradigm that exploits inference FLOPs bubbles to schedule pruning operations, thereby reducing additional overhead.

## 4.1. Environment-aware Adaptive Pruning

### 4.1.1. LIGHTWEIGHT ENVIRONMENT-AWARE SPARSITY VARIATIONS PREDICTOR

The optimal sparsity patterns of VLA models evolve dynamically during execution. To capture these changes efficiently, we introduce a lightweight environment-aware sparsity-pattern variation predictor. This module leverages visual feature similarities alongside a temporal context-conditioned trigger, enabling rapid and robust identification of changes.

**Lightweight Visual Similarity Metric.** To avoid introducing substantial computational overhead, we discard the popular attention-based semantic similarity (Xu et al., 2026), opting instead to leverage the visual features extracted by the VLA visual encoder to compute the similarity between step $t$ and $t-1$. Let $f_t, f_{t-1} \in \mathbb{R}^{N \times D}$ denote the image token features, where $N$ is the number of visual tokens and $D$ is the feature dimension. We define the similarity score as the average token-wise cosine similarity:

$$s_t = \frac{1}{N} \sum_{i=1}^{N} \frac{f_t^i \cdot f_{t-1}^i}{\|f_t^i\|_2 \|f_{t-1}^i\|_2} \tag{5}$$

We posit that if the visual features $f_t$ and $f_{t-1}$ exhibit high similarity, the sparsity pattern remains stable between frames. Conversely, significant deviation in visual features implies considerable variation in the sparsity pattern.

**Temporal Context–Conditioned Sparsity Trigger.** In open-world robotic manipulation, the environment is dynamic, causing the distribution of visual feature similarities

to evolve over time. Since we rely on these similarities to update sparsity patterns, a static decision criterion becomes brittle under distribution shifts. Therefore, we introduce a lightweight Temporal Context-Conditioned Sparsity Trigger, which leverages temporal context to adapt to such shifts. Specifically, we maintain a fixed-size sliding window $\mathcal{H}_t$ storing the similarities of the recent $T$ frames:

$$\mathcal{H}_t = \{s_{t-T}, s_{t-T+1}, \ldots, s_{t-1}\} \tag{6}$$

Based on this temporal context, we adopt a dynamic decision criterion where the sparsity update is triggered if the current similarity drops below the $p$-th quantile of the $\mathcal{H}_t$. The sparsity pattern update policy is formally defined as:

$$u_t = \mathbb{I}\left(s_t < \text{Quantile}(\mathcal{H}_t, p)\right) \tag{7}$$

Here, $p$ serves as a sensitivity hyperparameter, where a higher $p$ enhances responsiveness to subtle changes, whereas lower $p$ prioritizes stability. This mechanism yields self-regulative behavior. During rapid motion, the quantile naturally decreases, suppressing excessive updates to ensure stability. Conversely, in stable phases, the quantile increases, facilitating the sensitive detection of fine-grained variations.

### 4.1.2. TEMPORAL CONSISTENCY PRUNING

This section details the sparsity pattern computation. Upon triggering a sparsity update at frame $t$, we execute a dense inference to perform the pruning calculation. We first compute the instantaneous features from the current input at frame $t$, which are subsequently aggregated with historical features. The newly computed sparsity pattern is then applied starting from the frame $t+1$ for sparse inference.

To formalize this computation, we denote the update triggered at frame $t$ as the $\tau$-th sparsity update step. Specifically, as the current input reaches block $l$, following the formulation in Section 3.2, we can compute the current intermediate hidden states $\mathbf{X}^{l,int,\tau} = \mathcal{T}^l(\text{LN}(\mathbf{X}^{l,\tau}))$. Given that VLA models operate on single-sample streaming inputs, the dimensionality of $\mathbf{X}^{l,\text{int},\tau}$ is defined as $(1, S, C_{in})$. For notational simplicity, we omit the batch dimension in the subsequent formulation. To quantify the activation of the current input across structured channels, we compress the activation along the sequence dimension to obtain the instantaneous feature. Formally, for the $k$-th structured channel, the calculation is performed as follows:

$$\epsilon_k^{l,\tau} = \sum_{j=1}^{S} \left(\mathbf{X}_{j,k}^{l,\text{int},\tau}\right)^2 = \left\|\mathbf{X}_{:,k}^{l,\text{int},\tau}\right\|_2^2 \tag{8}$$

where $\epsilon_k^{l,\tau}$ represents the instantaneous feature of the $k$-th structured channel given the current input.

However, relying solely on instantaneous features is suboptimal, as the physical execution of VLA models exhibits

inherent temporal consistency. This temporal consistency is characterized by smooth, continuous transitions in both physical environments and proprioceptive states across adjacent frames, rather than discrete, abrupt jumps. In light of this, we aggregate the instantaneous feature with the historical feature. Specifically, we initialize the historical feature $\mathcal{E}^{l,(0)} \in \mathbb{R}^{C_{in}}$ for each block $l$ by averaging the instantaneous features computed by Equation 8 over the calibration dataset. For each subsequent update step $\tau$, we can compute the fused feature by leveraging the previous historical feature $\mathcal{E}^{l,(\tau-1)}$ as a temporal prior:

$$\tilde{\mathcal{E}}_k^{l,\tau} = \alpha \cdot \mathcal{E}_k^{l,(\tau-1)} + (1-\alpha) \cdot \epsilon_k^{l,\tau} \quad (9)$$

where $\alpha \in [0, 1]$ is the temporal inertia parameter. A larger $\alpha$ leads to a more conservative update. We update the historical feature on full channels using an exponential moving average. Here, the momentum $\lambda$ ensures temporal consistency, providing a stable prior for subsequent steps:

$$\mathcal{E}^{l,\tau} = \lambda \cdot \mathcal{E}^{l,(\tau-1)} + (1-\lambda) \cdot \epsilon^{l,\tau} \quad (10)$$

Finally, we can compute the sparsity pattern. Following PPsp (Le et al., 2025), we evaluate the significance of the $k$-th channel in the $l$-th layer based on both weight magnitude and fused feature. Let $\mathbf{W}^{l,\text{final}} \in \mathbb{R}^{C_{out} \times C_{in}}$ denote the final weight matrix in block $l$. The importance score $\mathcal{S}_k^{l,\tau}$ is formulated as:

$$\mathcal{S}_k^{l,\tau} = \left\| \left\{ \left| W_{i,k}^{l,\text{final}} \right|^2 \cdot \tilde{\mathcal{E}}_k^{l,\tau} \right\}_{i=1}^{C_{out}} \right\|_2 \quad (11)$$

where $\{\cdot\}$ denotes the set of elements, and $\mathcal{S}^{l,\tau} \in \mathbb{R}^{C_{in}}$. Crucially, identifying and pruning the $k$-th input channel of $\mathbf{W}^{l,\text{final}}$ (associated with low $\mathcal{S}_k^{l,\tau}$) necessitates the simultaneous removal of the corresponding $k$-th output channel of the intermediate transformations $\mathcal{T}^l$.

### 4.1.3. ANALYSIS OF COMPUTATIONAL COST

The computational overhead of EAP is primarily divided into visual feature similarity and sparsity pattern computation. For an image represented by $N$ visual tokens with feature dimension $D$ and a block with $C_{in}$ hidden dimensions and $C_{out}$ output channels, the sparsity pattern overhead includes instantaneous feature computation, feature fusion, historical feature updates, and importance score computation. Although the computational cost can be reduced through pre-computation, the worst-case complexity occurs when weight magnitudes are computed online, leading to the following FLOPs formulation:

$$\text{FLOPs} \approx 6ND + 2SC_{in} + 4C_{in}C_{out} + 6C_{in} \quad (12)$$

The computational overhead of EAP is marginal, as it primarily consists of element-wise operations that exhibit a negligible footprint compared to the VLA.

### 4.2. Interleaved Inference Orchestration

In this section, we first analyze the computational characteristics of VLA inference, revealing FLOPs bubbles arising from resource under-utilization. We then introduce **I**nterleaved **I**nference **O**rchestration (**I²O**), which exploits the complementary resource profiles between the VLM Backbone and Action Expert stages to absorb sparsity pattern computation into these bubbles with minimal overhead (Figure 2). Finally, we present hardware-efficient implementations to further accelerate both dense and sparse execution.

#### 4.2.1. FLOPs BUBBLES OF VLA INFERENCE

In this section, our profiling reveals temporal FLOPs bubbles during VLA inference due to a mismatch between model workload and hardware capacity, providing an opportunity to absorb sparsity pattern computation overhead.

**The VLM Backbone Stage.** This stage is dominated by large-scale General Matrix Multiplications (GEMMs) in transformer layers, rendering it inherently compute-bound (Williams et al., 2009). Given the massive model dimensions $(N, K)$ and the extensive sequence length $(M)$, the arithmetic intensity $(I)$ of these operations significantly exceeds the hardware's compute-to-memory ratio:

$$\frac{\text{FLOPs}}{\text{Bytes}} = \frac{2 \cdot M \cdot N \cdot K}{2 \cdot (NK + MK + MN)} \gg \frac{T_{\text{compute}}}{T_{\text{bandwidth}}} \quad (13)$$

where $T_{\text{compute}}$ and $T_{\text{bandwidth}}$ compute throughput and memory bandwidth, respectively. Consequently, the GPU Tensor Cores operate at near-peak saturation, while the memory bandwidth remains largely under-saturated, providing the necessary headroom for concurrent auxiliary tasks.

**The Action Expert Stage.** In stark contrast, this stage consists of lightweight MLPs or diffusion-based denoising steps (Ma et al., 2025). Operating under the real-time, robotic control streaming (batch size = 1), computational demand falls orders of magnitude below the GPU's peak parallel processing capacity (Markidis et al., 2018). In this state, the powerful Tensor Cores remain largely under-utilized, leaving a substantial computational reservoir that can be reclaimed to execute auxiliary tasks.

#### 4.2.2. HIDING PRUNING OVERHEADS VIA I²O

The conventional synchronous pruning stream is serially scheduled either preceding or succeeding the main inference stream, thereby linearly increasing the total latency. To circumvent this bottleneck, EcoVLA introduces **I**nterleaved **I**nference **O**rchestration (**I²O**) as illustrated in Figure 2, whose core philosophy is to interleave sparsity pattern computation into the FLOPs bubbles of the VLA pipelines.

Specifically, we decouple the sparsity pattern computation for step $t + 1$ from the main dense inference of the current

step $t$ by dispatching it to a parallel pruning stream. During the VLM Backbone stage, where the GPU is compute-saturated but memory bandwidth remains undersaturated, the pruning stream concurrently buffers the requisite intermediate activations. Subsequently, as the inference transitions to the Action Expert stage, I²O interleaves the sparsity pattern computation into FLOPs bubbles, effectively utilizing the idle Tensor Cores. As a result, I²O fully taps into the GPU's untapped computational potential, achieving a balanced workload distribution across the inference pipeline. By interleaving the sparsity pattern computation into the FLOPs bubbles, our approach avoids intense GPU resource contention, enabling low-latency sparsity-pattern computation with minimal overhead.

Building on this efficient orchestration, we now turn our attention to the latency analysis of I²O. Let $T_{infer}$ denote the VLA inference latency and $T_{prune}$ the overhead. In conventional synchronous approaches, the total latency is additive: $L_{synch} = T_{infer} + T_{prune}$. In I²O, the latency becomes $L_{I^2O} = T_{infer} + \delta$, where $\delta$ accounts for the overhead induced by concurrent execution, such as Streaming Multiprocessor (SM) scheduling costs, memory bandwidth contention and minor GPU resource competition. Owing to the lightweight design of the adaptive pruning module and its execution within FLOPs bubbles, $\delta$ is minimal. Consequently, this orchestration ensures that adaptive pruning is integrated seamlessly without impacting the latency-sensitive VLA control loop, thus maintaining the high-frequency reactivity essential for smooth robotic manipulation.

### 4.3. Hardware-efficient Implementation

In this section, we detail the hardware-level optimizations implemented for both dense and sparse VLA inference to achieve practical inference acceleration.

#### 4.3.1. SPARSE EFFICIENT KERNELS

We apply three kernel-level optimizations targeting memory efficiency and computation throughput in sparse inference.

**Sparse Linear Transformation Kernel.** In the standard PyTorch implementation, the retained weights are indexed before performing the linear transformation, which incurs additional memory I/O overhead. In contrast, we have implemented a sparse linear transformation Triton kernel (Tillet et al., 2019; Liu et al., 2023b) that directly loads only the retained weights during computation, avoiding this overhead and improving efficiency.

**Memory coalescing.** As described in Section 3.2, when performing pruning on the $W_{\text{down}}$ weights of the MLP, we actually prune the columns of the weights. In the sparse implementation, this results in indices pointing to non-contiguous memory regions, leading to reduced memory access effi-

ciency (Yang et al., 2018). We simply store these matrices in column-major format to enhance memory access locality. Similarly, we also store the output projection weights $W_O$ in the self-attention module in a column-major format.

**High-Performance Fused Kernels.** During sparse VLA inference, the MLP computes gate projection, up projection, SiLU activation, and element-wise multiplication. We fuse these four operations into a single Triton kernel that reads the input once and keeps intermediate results in registers, eliminating redundant memory traffic and reducing kernel launches from 4 to 1, yielding nearly $2\times$ speedup in practice.

#### 4.3.2. DENSE METRIC ACCELERATION

In addition to I²O, we apply two memory-level optimizations to accelerate sparsity pattern computation.

**Allocation-Free Caching.** As noted in Equation 11, computing the sparsity pattern involves L2 norms of squared weights and activation norms, where standard implementations suffer from repeated weight loading and dynamic memory allocation. We address this by pre-computing L2 norms of squared weights as compact vectors (a 99.97% reduction) and pre-allocating static activation buffers, transforming memory-bound operations into lightweight lookups while eliminating runtime allocations.

**Batched Metric Computation.** Per-layer sparsity computation incurs many separate kernel launches. We instead stack L2 norms of squared weights and activation buffers across layers into contiguous tensors, enabling kernel fusion that amortizes launch overhead and exposes layer-level parallelism.

## 5. Experiments

### 5.1. Experimental Settings

**Baselines.** To validate the generalizability of EcoVLA, we evaluate it across diverse VLA architectures. In simulation, we conduct experiments on open-source VLA models, including OpenVLA-OFT (Kim et al., 2025), $\pi_{0.5}$ (Intelligence et al., 2025), and CogACT (Li et al., 2024a). For fair comparison, we benchmark EcoVLA against Wanda (Sun et al., 2023), a mainstream static pruning method. Beyond standalone improvements, we demonstrate that EcoVLA is broadly compatible and can be stacked with other acceleration techniques. Experiments combining EcoVLA with FastV (Chen et al., 2024) and VLA-Cache (Xu et al., 2026) yield substantial additional speedups with negligible performance degradation. For real-world evaluation, we fine-tune $\pi_{0.5}$ for deployment on a 7-DoF Kinova Gen3 robotic arm and a 19-DoF Franka equipped with XHand. See Appendix A for further details.

**Evaluation Protocol.** To ensure a comprehensive compar-

*Table 1.* Performance of EcoVLA on OpenVLA-OFT in LIBERO at 25% and 40% pruning ratios. For VLA-Cache-based methods, speedup is computed using the eager `LlamaAttention` latency reported in parentheses.

| Method | Success Rate (%) ↑ | | | | | FLOPs (T) ↓ | Latency (ms) ↓ | Speedup ↑ |
|---|---|---|---|---|---|---|---|---|
| | LIBERO-Spatial | LIBERO-Object | LIBERO-Goal | LIBERO-Long | Average | | | |
| Vanilla | 97.6 | 98.4 | 96.2 | 94.6 | 96.7 | 4.05 (100.0%) | 143.56 (162.78) | 1.00× |
| FastV | 96.4 | 97.2 | 89.0 | 94.8 | 94.4 | 2.49 (61.48%) | 118.57 | 1.21× |
| VLA-Cache | 96.6 | 98.6 | 96.4 | 94.8 | 96.6 | 3.03 (74.81%) | 148.51 | 1.10× |
| *Pruning Ratio 25%* | | | | | | | | |
| Wanda | 95.8 | 98.8 | 87.2 | 93.2 | 93.8 | 3.14 (77.53%) | 124.32 | 1.15× |
| **Ours** | 97.4 | 98.8 | 94.6 | 96.4 | 96.8 | 3.23 (79.75%) | 113.98 | 1.26× |
| **FastV + Ours** | 96.6 | 98.2 | 94.0 | 96.0 | 96.2 | 1.96 (48.39%) | 65.85 | 2.18× |
| **VLA-Cache + Ours** | 96.8 | 98.4 | 92.4 | 94.2 | 95.5 | 2.43 (60.00%) | 121.24 | 1.34× |
| *Pruning Ratio 40%* | | | | | | | | |
| Wanda | 89.2 | 98.4 | 77.0 | 90.6 | 88.8 | 2.57 (63.46%) | 106.47 | 1.35× |
| **Ours** | 95.6 | 97.8 | 89.4 | 93.2 | 94.0 | 2.74 (67.65%) | 101.58 | 1.41× |
| **FastV + Ours** | 94.4 | 97.8 | 85.8 | 93.4 | 92.9 | 1.64 (40.49%) | 61.16 | 2.35× |
| **VLA-Cache + Ours** | 93.0 | 98.2 | 89.6 | 93.4 | 93.6 | 2.03 (50.12%) | 108.48 | 1.50× |

*Table 2.* Performance of EcoVLA on $\pi_{0.5}$ in LIBERO at 25% and 37.5% pruning ratios.

| Method | Sparsity | Success Rate (%) ↑ | | | | | FLOPs (T) ↓ | Latency (ms) ↓ | Speedup ↑ |
|---|---|---|---|---|---|---|---|---|---|
| | | LIBERO-Spatial | LIBERO-Object | LIBERO-Goal | LIBERO-Long | Average | | | |
| Vanilla | 0% | 98.8 | 98.2 | 98.0 | 92.4 | 96.9 | 1.99 (100.0%) | 81.94 | 1.00× |
| **Ours** | 25.0% | 98.2 | 98.6 | 98.4 | 91.6 | 96.7 | 1.64 (82.41%) | 62.66 | 1.31× |
| | 37.5% | 97.8 | 98.4 | 96.8 | 87.0 | 95.0 | 1.47 (73.87%) | 55.98 | 1.46× |

*Table 3.* Performance of EcoVLA on CogACT in SIMPLER at 25% and 40% pruning ratios.

| SIMPLER | Method | Sparsity | Success Rate (%) ↑ | | | | | FLOPs (T) ↓ | Latency (ms) ↓ | Speedup ↑ |
|---|---|---|---|---|---|---|---|---|---|---|
| | | | Pick Coke | Move Near | Open/Close | Open Top | Average | | | |
| Visual Matching | Vanilla | 0 | 93.3 | 83.8 | 74.5 | 41.7 | 73.3 | 1.81 (100.0%) | 104.16 | 1.00× |
| | **Ours** | 25% | 95.0 | 82.1 | 70.8 | 38.9 | 71.7 | 1.45 (80.11%) | 72.65 | 1.43× |
| | **Ours** | 40% | 93.0 | 85.4 | 73.5 | 42.6 | 73.6 | 1.25 (69.06%) | 66.43 | 1.57× |
| Variant Aggregation | Vanilla | 0 | 88.7 | 76.8 | 26.7 | 51.9 | 61.0 | 1.81 (100.0%) | 105.87 | 1.00× |
| | **Ours** | 25% | 85.9 | 75.3 | 27.2 | 46.0 | 58.6 | 1.47 (81.22%) | 73.98 | 1.43× |
| | **Ours** | 40% | 86.1 | 74.3 | 33.1 | 48.7 | 60.6 | 1.28 (70.72%) | 66.25 | 1.60× |

*Table 4.* Performance of EcoVLA on $\pi_{0.5}$ on a real-world robot.

| Method | Task1 | Task2 | Task3 | Latency (ms) |
|---|---|---|---|---|
| Baseline | 12/20 | 18/20 | 16/20 | 86.08 |
| Ours | 12/20 | 16/20 | 15/20 | 68.40 |

*Table 5.* Overhead of Pruning Stream.

| Execution Method | Latency (ms) |
|---|---|
| Normal VLA Inference | 143.56 |
| I²O | 148.06 |

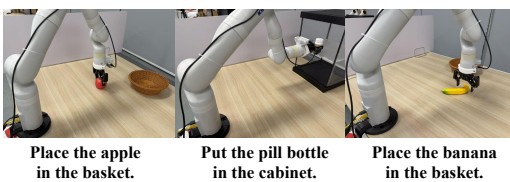

**Place the apple in the basket.** **Put the pill bottle in the cabinet.** **Place the banana in the basket.**

*Figure 3.* Robot Manipulation on Kinova Gen3 Platform.

on an NVIDIA Jetson Orin. Latency is measured following VLA-Cache. For VLA-Cache-based experiments, we use eager `LlamaAttention` (Touvron et al., 2023a;b) instead of `FlashAttention` (Dao et al., 2022); otherwise, `FlashAttention` is enabled by default. Hyperparameter details are provided in the Appendix A.3.

**Benchmarks.** We evaluate EcoVLA on two simulators and real-robot tasks. The LIBERO (Liu et al., 2023a) benchmark is designed to evaluate robotic manipulation capabilities across four task suites: LIBERO-Spatial, LIBERO-Object, LIBERO-Goal, and LIBERO-Long. Each task suite examines different capabilities of VLA. The SIMPLER (Li et al., 2024b) simulation environment provides two evaluation settings, including Visual Matching and Variant Aggregation. We evaluate four tasks on the Google Robot arm: 1)

ison, we adopt different pruning ratios. Specifically, for OpenVLA-OFT (Kim et al., 2025) and CogACT (Li et al., 2024a), we perform evaluations at pruning ratios of 25% and 40%, while for $\pi_{0.5}$ (Intelligence et al., 2025), we evaluate at 25% and 37.5% to accommodate the architecture. The evaluation metrics primarily include task success rate(%), inference latency(ms), and FLOPs(T).

**Implementation Details.** We follow the original settings of FastV (Chen et al., 2024) and VLA-Cache (Xu et al., 2026). Unless otherwise specified, experiments run on an NVIDIA RTX 3090 GPU, while edge-device results are obtained

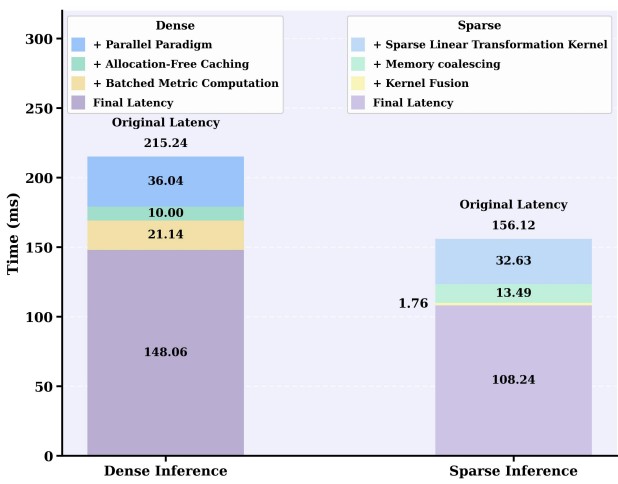

*Figure 4.* Acceleration breakdown for dense and sparse inference.

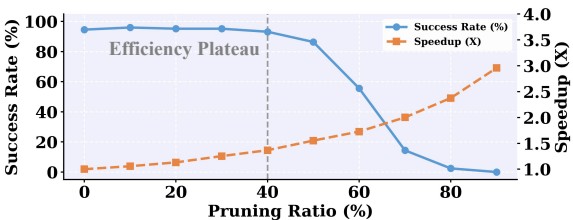

*Figure 5.* Trade-off between Success Rate and Latency.

*Pick Coke Can (PickCan)*, 2) *Move Near (MoveNear)*, 3) *Open/Close Drawer (Drawer)*, and 4) *Open Top Drawer and Place Apple (DrawerApple)*. To assess EcoVLA in real-world settings, we evaluate it on two real-robot platforms, each covering three tasks, as shown in Figure 3 and Figure 7. More details are provided in the Appendix B.1.

### 5.2. Main Results

**Results on OpenVLA-OFT.** Using the LIBERO benchmark, we evaluate EcoVLA on OpenVLA-OFT and show its compatibility by combining it orthogonally with acceleration techniques like FastV and VLA-Cache. Table 1 shows that EcoVLA achieves $1.26\times$ and $1.41\times$ speedups at $25\%$ and $40\%$ pruning ratios, with comparable performance at $25\%$ sparsity and only a $2.7\%$ drop at $40\%$ sparsity. The robustness of EcoVLA is particularly pronounced on the pruning-sensitive LIBERO-Goal benchmark, significantly outperforming the mainstream method Wanda. Specifically, EcoVLA establishes a commanding lead of $7.4\%$ and $12.4\%$ over Wanda at the respective pruning ratios. Beyond this robustness, EcoVLA achieves lower latency than Wanda despite an additional pruning overhead, as $I^2O$ conceals the pruning cost within FLOPs bubbles. Combined with our hardware-efficient design, this enables EcoVLA to deliver distinctly higher wall-clock speedups over Wanda.

Our results empirically validate the broad compatibility of EcoVLA. When combined with FastV ($50\%$ token pruning),

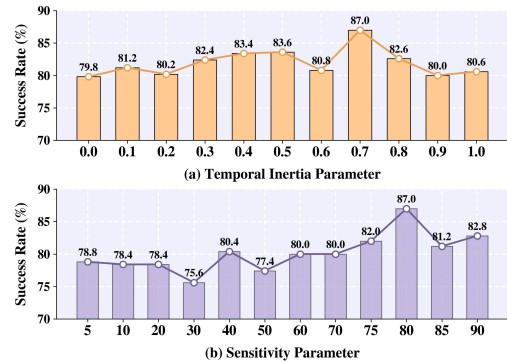

*Figure 6.* Impact of Hyperparameters $\alpha$ and $p$.

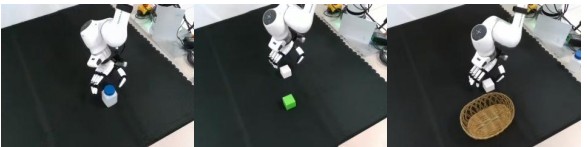

*Figure 7.* Experiments on Franka Research 3 robot with XHand.

it achieves a $2.18\times$ speedup with a negligible $0.5\%$ loss at $25\%$ sparsity, and reaches a $2.35\times$ speedup at $40\%$. When combined with VLA-Cache, it achieves a $1.34\times$ speedup with a negligible $1.2\%$ loss at $25\%$ sparsity, and reaches a $1.5\times$ speedup at $40\%$. Notably, these combinations deliver significantly higher speedups compared to the standalone baselines. While generally incurring only a slight decline in success rate, intriguingly, we observe a performance improvement in the FastV combination. We attribute this phenomenon to the regularization effect induced by our precise, adaptive model sparsification (Jin et al., 2022).

**Results on $\pi_{0.5}$.** To demonstrate the cross-model generalizability of EcoVLA, we conduct experiments on the state-of-the-art VLA model, $\pi_{0.5}$, as shown in Table 2. At $25\%$ and $37.5\%$ pruning ratios, EcoVLA achieves $1.31\times$ and $1.46\times$ speedups with marginal accuracy drops of $0.2\%$ and $1.9\%$, respectively. Notably, for the LIBERO-Object, we observe a $0.2\%$ accuracy improvement at the $37.5\%$ pruning ratio. This observation corroborates the regularization benefit discussed earlier, suggesting that selective pruning effectively filters out noise from redundant parameters. Most significantly, despite $\pi_{0.5}$'s inherently efficient structure and low latency ($81.94$ ms), EcoVLA still manages to extract a substantial $1.46\times$ acceleration. This capability to further accelerate an already fast model redefines the standards for efficient real-time deployment.

**Results on CogACT.** We evaluate EcoVLA's generalization on CogACT within SIMPLER, as summarized in Table 3. In the Visual Matching setting, EcoVLA achieves a $1.57\times$ speedup at a $40\%$ pruning ratio with a $0.3\%$ improvement in success rate. Similarly, in the Variant Aggregation setting, it reaches a $1.6\times$ speedup at $40\%$ pruning while incurring a marginal accuracy reduction of $0.4\%$.

*Table 6.* Additional results on a Franka Research 3 robot with XHand.

| Method | Success Rate ↑ | | | Latency (ms) ↓ |
|---|---|---|---|---|
| | pickup reagent bottle | stack white block on green | put block in the basket | |
| $\pi_{0.5}$ | 27/50 | 23/50 | 25/50 | 81.81 |
| Ours (25%) | 26/50 | 22/50 | 25/50 | 62.53 |

*Table 7.* Performance of EcoVLA under noise.

| Noise | Clean | Visual Noise | Lighting Variation | Jitter | Occlusion |
|---|---|---|---|---|---|
| Ours (25%) | 26/50 | 25/50 | 24/50 | 27/50 | 21/50 |

*Table 8.* Activation shift between the first and middle frames under different pruning ratios.

| Sparsity | Attention Overlap | Average Replaced Important Heads | MLP Overlap | Average Replaced Important Channels |
|---|---|---|---|---|
| 25% | 0.95 | 1.3/24 | 0.86 | 1159/8256 |
| 40% | 0.92 | 1.6/20 | 0.78 | 1427/6605 |

*Table 9.* Results on the Orin under different pruning ratios.

| Hardware | Sparsity | Latency (ms) ↓ | Speedup ↑ |
|---|---|---|---|
| Orin | 0% | 702.26 | 1.00× |
| Orin | 25% | 548.67 | 1.28× |
| Orin | 40% | 491.23 | 1.43× |

**Results on Real Robot.** We evaluate real-world performance on a physical 7-DoF Kinova Gen3 arm controlled by $\pi_{0.5}$. Figure 3 illustrates robot manipulation tasks performed during the experiments. As shown in Table 4, our approach delivers a 1.26× wall-clock speedup at the cost of minor performance loss, underscoring its viability for real-robot deployment. Further analysis is provided in Appendix B.2.

### 5.3. More Results

**Ablation Study of Acceleration on OpenVLA-OFT.** Figure 4 details the acceleration breakdown. For dense inference, bubble scheduling, buffer optimization, and batched metrics reduce latency by 36.04 ms, 10.00 ms, and 21.14 ms respectively, yielding 148.06 ms. For sparse inference, utilizing sparse kernels, memory coalescing, and kernel fusion cuts latency by 32.63 ms, 13.49 ms, and 1.76 ms, resulting in 108.24 ms.

**Trade-off between Performance and Latency.** As shown in Figure 5, we analyze the trade-off between success rate and latency on LIBERO-Long. While the success rate remains robust below 40% pruning, it deteriorates rapidly beyond this threshold. We observe an optimal trade-off between performance and latency at a 40% pruning ratio.

**Overhead of Pruning Stream on OpenVLA-OFT.** The parallel pruning stream introduces overhead $\delta$, detailed in Table 5. As analyzed in Section 4.2.2, owing to the lightweight design of the EAP and its execution within FLOPs bubbles, this overhead is minimal, limited to 4.5 ms.

**Hyperparameter Studies.** Figure 6 analyzes the impact of $\alpha$ and $p$ on LIBERO-Long. High $\alpha$ over-relies on historical features, causing sparsity lag, while low $\alpha$ leads to instability. Performance peaks at $\alpha = 0.7$, validating our design. Similarly, low $p$ overlooks subtle cues, whereas high $p$ causes hypersensitivity to noise. The peak success rate at $p = 80\%$ confirms EcoVLA effectively filters noise while maintaining the sensitivity required for robust operation. Additional analysis is provided in Appendix C.

**Additional Real-Robot Results and Noise Robustness.** To assess EcoVLA's robustness under noise, we conduct additional real-robot experiments. As shown in Figure 7, we evaluate real-world performance on a Franka Research 3 robot with an XHand, across three tasks. Table 6 shows that EcoVLA achieves a 1.31× speedup with minimal reduction in success rate. We further introduce four perturbation types to the pickup reagent bottle task to simulate visual noise. Table 7 shows that EcoVLA remains relatively robust under visual noise, lighting variation, and jitter, while occlusion introduces a more visible degradation.

**Activation Shift Analysis.** Figure 1 shows that channel importance varies with environmental changes during VLA execution. To further quantify this observation, Table 8 reports the activation shift between important channels selected from different visual frames on LIBERO. The results show that important attention heads and MLP channels change over time, especially at higher sparsity levels, supporting the need for environment-aware adaptive pruning. The metric definition is provided in Appendix D.

**Results on Edge Device.** To evaluate the edge-device applicability of EcoVLA, we further conduct experiments on the NVIDIA Jetson Orin platform. Table 9 shows that EcoVLA still achieves substantial speedup, demonstrating its effectiveness for resource-constrained edge deployment. Detailed hardware specifications are provided in Appendix E.

## 6. Conclusion

We present EcoVLA, a training-free, plug-and-play adaptive pruning framework for VLA models that combines Environment-aware Adaptive Pruning (EAP) with Interleaved Inference Orchestration ($I^2O$) to update sparsity online with negligible overhead. We validate EcoVLA on both real robots and high-fidelity simulators.

## Acknowledgements

This work was supported by the Fundamental and Interdisciplinary Disciplines Breakthrough Plan of the Ministry of Education of China (No. JYB2025XDXM113) and the National Natural Science Foundation of China (No. 62332016).

## Impact Statement

This paper presents work whose goal is to advance the field of machine learning. There are many potential societal consequences of our work, none of which we feel must be specifically highlighted here.

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

# A. Experimental Settings

## A.1. VLA Model Details

**OpenVLA-OFT.** Derived from the OpenVLA architecture, OpenVLA-OFT incorporates an Optimized Fine-Tuning strategy that simultaneously enhances manipulation precision and computational efficiency. By adopting a continuous action space optimized via $L1$ regression alongside parallel decoding and action chunking mechanisms, it achieves superior throughput and success rates on the LIBERO benchmark. A key characteristic of this variant is the deployment of bidirectional attention during inference.

$\pi_{0.5}$. $\pi_{0.5}$ is a generalist Vision-Language-Action (VLA) model designed to achieve broad open-world generalization in mobile manipulation by leveraging a heterogeneous co-training recipe. Building upon the $\pi_0$ architecture and the PaliGemma VLM backbone, $\pi_{0.5}$ integrates diverse data sources—including cross-embodiment robot data, multimodal web data, and high-level semantic predictions—to bridge the generalization gap. The model employs a hierarchical inference mechanism where it first predicts a high-level semantic subtask (e.g., "pick up the pillow") based on the visual observation and global instruction, which then conditions a specialized "action expert" to generate continuous low-level control actions via flow matching. This capability is developed through a two-stage training process that transitions from scalable discrete token pre-training to precise continuous flow-matching post-training, enabling the execution of complex, long-horizon tasks in environments completely unseen during training.

**CogACT.** This architecture synthesizes perception and reasoning by employing DINOv2 and SigLIP for visual encoding alongside a LLaMA2-7B language backbone. To bridge the gap between high-level cognition and low-level action, CogACT utilizes a specialized Diffusion Transformer (DiT). By conditioning this diffusion-based action module on the features extracted by the VLM, the model effectively addresses the challenges of generating precise, continuous, and temporally correlated robotic trajectories.

## A.2. Acceleration Method Details

**FastV.** FastV addresses inference latency in Large Vision-Language Models (LVLMs) by mitigating visual token redundancy. The method is grounded in the observation that deep layers often exhibit an attention sink phenomenon, where visual tokens consume substantial computational resources despite receiving minimal attention weights. To counter this, FastV introduces a plug-and-play mechanism that monitors attention scores to dynamically discard low-utility visual tokens after a certain depth, thereby reducing FLOPs while preserving model accuracy.

**VLA-Cache.** VLA-Cache is a training-free inference accelerator tailored for robotic VLA models. It exploits the observation that visual scenes in robotic tasks remain largely stable between consecutive frames, particularly in background areas. By distinguishing between static and dynamic elements, the method recycles KV-cache states for unchanged tokens while enforcing full computation for critical, task-relevant features to ensure precision. Furthermore, it incorporates an adaptive strategy that modulates the caching ratio according to layer-specific attention patterns.

## A.3. Implementation Details

For OpenVLA-OFT, we set the sensitivity parameter $p = 5\%$. Regarding temporal inertia parameter, we set $\alpha = 0.7$ for LIBERO-Spatial and LIBERO-Long to reduce historical reliance in dynamic spatial layouts. Conversely, for LIBERO-Object and LIBERO-Goal, we use a higher $\alpha = 0.9$ as these manipulation tasks feature relatively stable environments. For $\pi_{0.5}$, given its enhanced execution stability, we set the sensitivity parameter to $80\%$. The temporal inertia parameter is kept fixed at 0.7. For CogACT, we set the sensitivity parameter $p = 5\%$, $\alpha = 0.7$. We set $\lambda$ to 0.99 in all experiments, following prior work (Le et al., 2025) and engineering-oriented empirical screening.

# B. Real-robot Setup and Analysis

## B.1. Real-robot Setup

We conduct real-robot experiments on two platforms: a Kinova Gen3 robotic arm and a Franka robot equipped with XHand. The corresponding experimental setups are shown in Figure 8 and Figure 9.

**Kinova Gen3**. We use two Intel RealSense D435i cameras: one providing a third-person view and the other mounted on the robot wrist. For each task, we collect 50 demonstrations. When fine-tuning the $\pi_{0.5}$ model, we freeze the VLM backbone

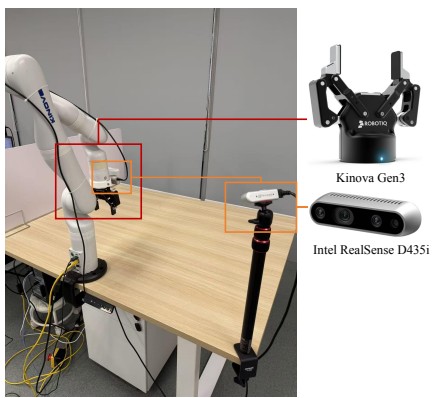

*Figure 8.* Kinova Gen3 robot setup.

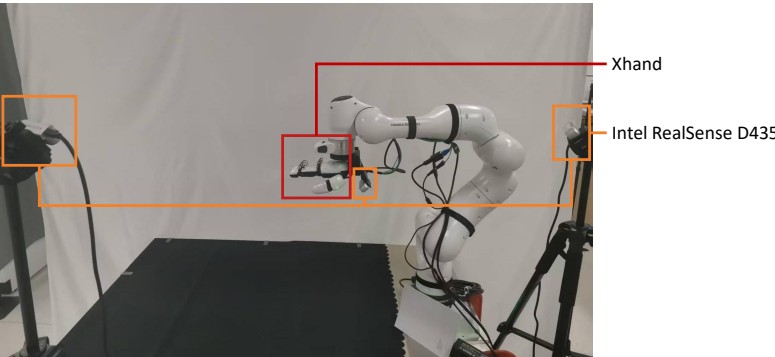

*Figure 9.* Franka equipped with XHand robot setup.

and apply LoRA-based adaptation, while fully fine-tuning the action expert. We use a global batch size of 32 and train the model for $10k$ steps on a single GPU.

**Franka equipped with XHand**. We use three Intel RealSense D435i cameras, including a front-left view, a rear-left view, and a wrist-mounted view. For each task, we collect 100 demonstrations. The fine-tuning configuration follows the same setting as described above.

### B.2. Analysis on Kinova Gen3

During evaluation, to more thoroughly assess the model's capability, we test on 20 randomly sampled object placements within the training range, re-randomizing the object position in every trial to probe the model's spatial reasoning and generalization across object locations. On real-robot tasks, while achieving a $1.26\times$ speedup, EcoVLA exhibits a minor performance drop compared to $\pi_{0.5}$. We further observe that most failures occur when the object is placed near the boundary of the workspace. We conjecture that such edge-case placements require finer-grained spatial cues and stricter geometric constraints, making the pruned model more susceptible to representation loss and thus less robust in spatial generalization.

## C. Hyperparameter Analysis

We evaluate the influence of sensitivity parameter $p$ and temporal inertia parameter, $\alpha$ on success rates using $\pi_{0.5}$.

**Impact of Sensitivity Parameter.** As shown in Figure 6 (b), we evaluate the impact of the sensitivity parameter $p$, as it directly dictates EcoVLA's responsiveness to environmental dynamics. At low $p$ values, EcoVLA adopts a conservative strategy, updating sparsity patterns only during drastic environmental changes, thereby overlooking subtle cues. Conversely, an excessively high $p$ induces hypersensitivity, where sensor noise or lighting fluctuations are misidentified as dynamics. This triggers redundant updates that disrupt temporal continuity. Performance peaks at $p = 80\%$, a setting that effectively filters noise while retaining sufficient sensitivity to capture physical transitions, ensuring robust operation in complex environments.

**Impact of Temporal Inertia Parameter.** We investigate the properties of $\alpha$ on LIBERO-Long (Figure 6 (a)), as this parameter is critical for long-horizon tasks with varying spatial layouts. High $\alpha$ excessively weights history, causing the model to miss new critical features, whereas extremely low $\alpha$ relies on single-frame inputs, disrupting inference stability. Performance peaks at $87.0\%$ with an intermediate $\alpha$, validating our Temporal Consistency Pruning strategy. This confirms that EcoVLA effectively filters high-frequency noise while maintaining sensitivity to environmental dynamics.

## D. Details of Activation Shift Analysis

We quantify activation shift by measuring the overlap between important channels selected from different visual frames. Let $C_t^{(K)}$ denote the set of top-$K$ important components at frame $t$, where the components can be either attention heads or MLP

channels. Given two frames $t_1$ and $t_2$, we define their overlap as

$$\text{Overlap}(t_1, t_2) = \frac{\left| C_{t_1}^{(K)} \cap C_{t_2}^{(K)} \right|}{K}. \tag{14}$$

A higher overlap indicates that the same components remain important across frames, while a lower overlap suggests stronger temporal activation shift. We additionally report the number of replaced important components, computed as

$$\text{Replaced}(t_1, t_2) = K - \left| C_{t_1}^{(K)} \cap C_{t_2}^{(K)} \right|. \tag{15}$$

In our analysis, we compare the important components identified from the first frame and the middle frame across different tasks, and report the average overlap rate. This provides a direct measurement of whether the important attention heads and MLP channels remain temporally stable during VLA inference.

## E. Detailed hardware specifications

We conduct edge-device experiments on an NVIDIA Jetson AGX Orin Developer Kit. The device runs Ubuntu 22.04.5 LTS with Linux kernel 5.15.148-tegra on an aarch64 platform. The L4T release is R36.4.3. The device is configured under the 50W power mode, and CUDA compilation tools are version 12.1.

*Table 10.* Hardware and software specifications of the edge device.

| Item | Specification |
| --- | --- |
| Device | NVIDIA Jetson AGX Orin Developer Kit |
| Architecture | aarch64 |
| Operating system | Ubuntu 22.04.5 LTS |
| Kernel | Linux 5.15.148-tegra |
| L4T version | R36.4.3 |
| Memory | 61 GiB |
| Power mode | 50W mode |
| CUDA version | 12.1, V12.1.66 |

