# OpenReview forum: "EcoVLA: Environment-Aware Adaptive Pruning with Interleaved Inference Orchestration for Vision-Language-Action Models"
_ICML.cc/2026/Conference — ICML 2026 spotlight_

### Official Review · Reviewer_2msh · 2026-02-25

**Soundness:** 3
**Presentation:** 3
**Significance:** 3
**Originality:** 3
**Overall Recommendation:** 4
**Confidence:** 4

**Summary:**

This work addresses the challenge of accelerating Vision–Language–Action (VLA) models for real-time robotic execution without retraining by exploiting the environment-dependent activation patterns in VLA backbones. We propose EcoVLA, a dynamic pruning framework that adaptively updates pruning masks through Environment-Aware Pruning (EAP) by leveraging visual-feature similarity and temporally smoothed activation statistics. To minimize overhead, the framework incorporates Input-to-Output (I2O) scheduling to hide pruning computation within the idle GPU cycles of the lightweight action expert. Across LIBERO, SIMPLER, and real-world robot experiments, EcoVLA achieves a 1.3–2.2× speedup while maintaining high execution fidelity with negligible performance degradation.

**Compliance With Llm Reviewing Policy:**

Affirmed.

**Final Justification:**

Thank you very much for the authors' response. In my view, current VLA acceleration strategies based on action chunking and asynchronous inference are already sufficient for practical real-robot deployment. By contrast, acceleration through model pruning often comes at the cost of degraded performance, especially when additional training is required. This makes such approaches less appealing in practice. In comparison, the method proposed in this paper is plug-and-play and does not require training, which gives it a clear advantage in deployability. This is also the main reason why I gave a positive initial score. Therefore, I recommend acceptance.

**Key Questions For Authors:**

1. How robust is the feature-similarity trigger under real visual noise (camera jitter, lighting change, fast object motion)? Have you evaluated such perturbations?

2. Is the pruning-frequency trigger reliable? Are there failure modes where overly frequent updates accumulate overhead, or overly sparse updates make the masks stale?

3. In TCP (Temporal Consistency Pruning), how is the EMA smoothing factor chosen? Have you conducted sensitivity studies across different smoothing coefficients?

4. Does the actual wall-clock acceleration match FLOPs reduction? Have you measured latency as a function of sparsity level to establish the real-world sparsity–latency curve?

**Limitations:**

The limitations section could be strengthened. In particular, the paper should explicitly acknowledge that (1) evaluation is limited to a small set of robotic tasks; (2) the method assumes stable visual encoders and may be sensitive to real-world perception noise; and (3) adaptive pruning may degrade safety if the mask update behaves unpredictably under distribution shift. A more explicit discussion would improve transparency.

**Strengths And Weaknesses:**

**Strengths:**

1. Clear and practical motivation: The main bottleneck of current VLAs lies in the large backbone, whose activation patterns vary significantly across task phases. Static pruning cannot adapt to such dynamics, making the problem highly relevant.

2. No training overhead and plug-and-play: EcoVLA requires no additional training, which substantially improves deployability.

3. Effective systems contribution via I2O: The insight that the VLM backbone is compute-bound while the action expert is under-utilized is well leveraged. Interleaving pruning updates inside these “pipeline bubbles” is an elegant engineering solution. The fact that EcoVLA works on top of existing accelerators (FastV, VLA-Cache) further demonstrates its modularity and practicality.

**Weaknesses:**

1. Limited methodological novelty; heavily engineering-oriented.
The core design essentially combines activation-based pruning with similarity-triggered updates and kernel-level optimizations. EAP’s trigger mechanism resembles prior activation-driven dynamic pruning methods, with visual feature similarity being a relatively minor adaptation.

2. Insufficient analysis of activation shift.
Although the paper motivates the method by showing activation drift across task phases, it does not provide a systematic characterization of such shifts or analyze differences across task modes. The argument is intuitive but not convincingly supported.

3. Lack of robustness evaluation for the pruning triggers.
The stability of the pruning masks under visual noise, occlusion, lighting variation, or camera jitter is not examined. It is unclear whether the similarity-based triggers remain reliable in realistic noisy conditions.

4. Thin real-robot validation.
Only a small number of pick-and-place tasks are included, which is insufficient to substantiate claims of broad deployability.

---

> ### Author Rebuttal · Authors · 2026-03-31
>
> # Response to Reviewer 2msh
> We sincerely appreciate the reviewer's insightful feedback. Unless otherwise specified, all rebuttal experiments are conducted on OpenVLA-OFT in LIBERO-Goal.
> ## W1: EcoVLA Novelty
> We thank the reviewer for recognizing the novelty and importance of our system design. We would like to clarify that our contribution is not a simple combination of pruning and system optimization, **but an algorithm-system co-design that makes adaptive pruning practical for real-time VLA inference. Our design is driven by the runtime characteristics of VLA models**. Starting from why existing LLM/VLM pruning methods do not transfer directly to VLA at either the algorithmic or system level, we design EcoVLA accordingly. On the algorithmic side, EAP is intentionally designed to be lightweight, where the visual similarity trigger together and temporal consistency pruning make **adaptive pruning streamlined yet effective while keeping the online overhead minimal**. More elaborate designs are often impractical in this setting due to runtime overhead. On the system side, $I^2O$ is essential for turning FLOPs reduction into actual wall-clock speedup. **To our knowledge, it's the first work on adaptive pruning for VLA models, and we therefore provide a full framework to make it practical for real deployment.**
> ## W2:Activation Shift
> We use activation shift as a motivating observation for EcoVLA, rather than as a standalone empirical contribution. This is consistent with prior work[1], which we will cite more clearly in the revision. Our current support mainly comes from the experiments showing that EcoVLA outperforms static pruning. Following the suggestion, we further analyze activation shift in VLA. Let $C_t^{(K)}$ denote the top-$K$ important channels at frame $t$, and define the overlap between $t_1$ and $t_2$ as $$\mathrm{Overlap}(t_1,t_2)=\frac{|C_{t_1}^{(K)} \cap C_{t_2}^{(K)}|}{K} $$ We compare the important channels of the first and middle frames across various tasks, and compute average overlap rate. The results show that important channels shift over time, directly supporting our motivation.
> |Sparsity|Attention overlap|Replaced important heads|MLP overlap|Replaced important channels|
> |-|-|-|-|-|
> |25%|0.95|1.3/24|0.86|1159/8256|
> |40%|0.92|1.6/20|0.78|1427/6605|
>
> [1] Liu Z, Wang J, Dao T, et al. Deja vu: Contextual sparsity for efficient llms at inference time
> ## W3,W4 and Q1: Robustness and Real-robot
> We agree that robustness under noisy conditions is important, **and this is explicitly considered in the design of EAP.** The temporal context-conditioned sparsity trigger avoids a static threshold by making decisions based on temporal context, reducing sensitivity to transient noise. Temporal consistency pruning aggregates instantaneous and historical features, improving stability under noise. More details are provided in Sec.4.1.1 and 4.1.2.
>
> To further evaluation, we also conduct real-robot experiments on Franka Research 3 with XHand([setup figure](https://anonymous.4open.science/r/anon-repo-9663/franka_xhand.pdf)), where noisy conditions are introduced through synthetic visual perturbations. Overall, EcoVLA maintains performance close to the baseline, and shows stable behavior under different visual perturbations.
>
> |Method|pickup reagent bottle|stack white block on green|put block in the basket|Latency|
> |-|-|-|-|-|
> |$\pi_{0.5}$|27/50|23/50|25/50|81.81|
> |Ours(25%)|26/50|22/50|25/50|62.53|
>
> |pickup reagent bottle|-|visual noise|lighting variation|jitter|occlusion|
> |-|-|-|-|-|-|
> |SR|26/50|25/50|24/50|27/50|21/50|
> ## Q2: Trigger Reliability
> Our temporal context-conditioned sparsity trigger is designed to balance these two failure modes. During rapidly changing phases, it suppresses excessive updates to maintain stability; in stable phases, it remains sensitive to fine-grained changes. The sensitivity parameter $p$ controls the trigger's sensitivity and affects the frequency as analyzed in Fig.6. On LIBERO, EcoVLA updates the sparsity pattern once every 6.94 inference steps on average.
> ## Q3: EMA Smoothing Factor
> We use a value selected through engineering-oriented testing and screening. In practice, we found that a relatively large value works better by preserving temporal consistency and reducing sensitivity to transient fluctuations, and therefore use. Our main hyperparameter analysis focuses on p and $\alpha$,the core EcoVLA parameters.
> |$\lambda$|SR|
> |-|-|
> |0.1|92.0|
> |0.3|92.4|
> |0.7|92.8|
> |0.9|93.4|
> |0.95|94.2|
> |0.99|94.6|
> ## Q4: Sparsity-Latency
> Yes, we have measured actual wall-clock latency at different sparsities. Tab.1-3 report both FLOPs reduction and wall-clock latency under multiple sparsities, and Fig.5 further shows the latency reduction trend with increasing sparsity, which shows that latency continues to decrease as sparsity increases, but with diminishing marginal returns. We believe this behavior is due to fixed system overheads, which become more prominent at higher sparsity.

---

> > ### Author Rebuttal · Reviewer_2msh · 2026-04-01
> >
> > Thank you to the authors for the detailed response and additional experiments. I no longer have concerns about the experimental evaluation. Regarding novelty, the rebuttal makes clear that the overall design is well motivated and tailored to the runtime constraints of VLA inference. However, the contribution still appears to me largely as an A+B style integration: a lightweight adaptive pruning strategy together with a system mechanism for translating theoretical computation savings into practical speedup. It remains unclear what fundamentally new technical insight is introduced by their combination beyond this integration.

---

> > > ### Author Response · Authors · 2026-04-02
> > >
> > > ## Response to Reviewer 2msh
> > > We thank the reviewer for the constructive follow-up and are pleased to know that the concerns regarding the experimental evaluation have now been resolved. We would like to clarify that both the research problem we study, namely adaptive pruning for closed-loop VLA inference, and the algorithm-system co-designed solution we propose, EcoVLA, have novelty that is not fully captured by viewing the paper as a simple A+B style combination.
> > >
> > > First, we would like to emphasize that adaptive pruning for closed-loop VLA inference is a previously unexplored setting with distinct constraints. EcoVLA is designed specifically for this setting, and, to the best of our knowledge, provides the **first** training-free, plug-and-play adaptive pruning framework for VLA that can adapt to environmental changes. **More broadly, many influential works derive their novelty not only from proposing an entirely standalone new algorithm , but also from identifying a previously unresolved bottleneck in an important new setting and redesigning the method and execution path accordingly [1,2,3].** We believe EcoVLA makes this type of contribution to real-time VLA inference.
> > >
> > > Specifically, the key challenges for VLA model pruning in real-time inference scenarios are twofold: 1. First, the sparsity patterns of VLA models change alongside environmental shifts; moreover, relying only on instantaneous observations fails to capture the temporal consistency of VLA model execution. 2. Second, adaptive pruning introduces online computational overhead. However, for VLA models that take single-batch inputs and are sensitive to inference frequency, this latency leads to stutter and jitter. **Consequently, adaptive pruning is practically unusable for real-time VLA control. Previously, no work has systematically analyzed or resolved this fundamental conflict.**
> > >
> > > **The fundamentally new technical insight is not the combination itself, but the observation that, in closed-loop VLA inference, adaptive pruning is only practically useful when the pruning policy and the execution path are co-designed under the same online constraint.** In this setting, a pruning method that is effective in isolation may still be too costly to run online, while system-level orchestration alone cannot ensure reliable closed-loop performance if the pruning policy itself is not architected for real-time dynamic adaptability. We view this feasibility condition as the key previously unexplored issue in adaptive pruning for VLA.
> > >
> > > To address this, we propose EcoVLA, a framework that tightly integrates algorithmic and system-level designs. At the algorithmic level, we design a lightweight yet effective EAP method, capable of capturing real-time changes in sparsity patterns in response to environmental shifts. Simultaneously, we introduce temporal consistency pruning to explicitly account for the continuity inherent in embodied interactions, thereby ensuring the stability of VLA execution. Systematically, we identify a unique runtime characteristic of VLA inference: a severe resource utilization imbalance between the heavy VLM backbone and the lightweight action expert. Based on this, we propose $I^2O$. By burying the pruning computations within the inherent FLOPs bubbles of VLA inference, $I^2O$ boosts hardware utilization and largely hides the pruning overhead, rendering online adaptive pruning highly practical for VLA models.
> > >
> > > Thus, rather than a naive A+B combination, these two modules form a tightly coupled architecture tailored for real-time VLA inference. EAP handles the real-time perception and precise computation of sparsity pattern shifts, whereas $I^2O$ maximizes the actual acceleration gained from the reduced FLOPs. Beyond this, we dedicated significant engineering effort to deliver a transferable, hardware-efficient implementation. **This provides the community with the first comprehensive training-free and plug-and-play framework in this domain, capable of rapid deployment across different hardware platforms.**
> > >
> > > ## Reference
> > > **[1] Dao T, Fu D, Ermon S, et al. Flashattention: Fast and memory-efficient exact attention with io-awareness**
> > >
> > > **[2] Kwon W, Li Z, Zhuang S, et al. Efficient memory management for large language model serving with pagedattention**
> > >
> > > **[3] Liu Z, Wang J, Dao T, et al. Deja vu: Contextual sparsity for efficient llms at inference time**

---

### Official Review · Reviewer_PWM5 · 2026-03-03

**Soundness:** 3
**Presentation:** 3
**Significance:** 3
**Originality:** 3
**Overall Recommendation:** 5
**Confidence:** 2

**Summary:**

The paper addresses the challenge of dynamic sparsity in VLA models, where optimal pruning patterns evolve based on the environment. Recognizing that adaptive pruning often introduces significant real-time overhead, the authors propose EcoVLA. This framework incorporates Environment-aware Adaptive Pruning and $I^2O$. Experimental results demonstrate that EcoVLA significantly enhances inference speeds across three distinct models and two benchmarks while maintaining performance.

**Compliance With Llm Reviewing Policy:**

Affirmed.

**Final Justification:**

The authors have addressed my primary concerns through the rebuttal. I remain positive about the paper and will maintain my current score.

**Key Questions For Authors:**

1. Is this approach transferable to unified multimodal models that do not utilize a standalone vision encoder?

2. The $I^2O$ module is demonstrated by exploiting FLOPs bubbles on an RTX 3090. Have the authors profiled this on edge devices? Do the same disparities exist on such hardware to allow for effectively "hiding" the pruning overhead?

3. While $I^2O$ successfully reduces latency, does the dual-stream orchestration significantly increase peak VRAM usage?

**Limitations:**

Yes

**Strengths And Weaknesses:**

### Strengths:
***Soundness***:
1. The authors provide a solid theoretical foundation by estimating the compute-to-memory ratio, which serves as a rigorous justification for the proposed $I^2O$ pipeline.

2. The evaluation includes comprehensive comparisons with existing state-of-the-art methods, specifically those integrated with FlashAttention, ensuring the results are benchmarked against competitive baselines.

3. The ablation studies offer an in-depth analysis of hyperparameter sensitivity ($\alpha$ and $p$), effectively illustrating the trade-off between success rate and latency.

4. The methodology is validated across three different models and two benchmarks, demonstrating robust performance.


***Presentation***:
1. The paper is well-structured and utilizes high-quality visualizations to clarify complex concepts effectively.

***Significance***:
1. The proposed method consistently outperforms baseline models like FastV and VLA-Cache in both execution speed and average success rate, which is a significant practical achievement.



***Originality***:
1. The approach is novel in its consideration of environmental changes and its use of a momentum-averaged feature update for sparsity patterns.

3. The dual-stream pipeline design, which interleaves sparsity pattern computation within computational "bubbles" (FLOPs bubbles) to hide overhead, is an original architectural contribution.



### Weaknesses:
***Soundness***:
1. The appendix notes that $p$ and $\alpha$ require task-specific tuning. This dependency challenges the "plug-and-play" claim, as users may need to perform significant calibration when deploying the model in new environments.



***Significance***:
1. The effectiveness of $I^2O$ in overlapping computation relies heavily on specific hardware characteristics. While results are shown for 3090 GPUs, it remains unclear how this method generalizes to edge devices (Jetson Orin), where the compute-bound vs. memory-bound disparities may differ, potentially diminishing the significance of the "hidden" pruning overhead.

---

> ### Author Rebuttal · Authors · 2026-03-31
>
> # Response to Reviewer PWM5
> We sincerely appreciate the reviewer's insightful feedback and positive comments on the methodology of our paper. Please find our responses below.
> ## W1: Clarifying Plug-and-Play
> We thank the reviewer for pointing this out. We agree that, as currently written, the paper may make the 'plug-and-play' claim sound stronger than intended. We mean that EcoVLA does not require additional training or fine-tuning before deployment. However, we agree that this should not be interpreted as requiring no deployment-time calibration under any new environment. However, since EcoVLA is training-free, and $p$ and $\alpha$ are lightweight deployment-time hyperparameters in our design with intuitive meanings, whose initial values can be roughly estimated from the task type. Concretely, $p$ controls the sensitivity of the update trigger, while $\alpha$ controls the temporal inertia in feature fusion. Therefore, such adjustment is low-cost in practice. We will revise the paper to clarify this distinction more explicitly: EcoVLA is training-free and plug-and-play in the sense of requiring no training, while transfer to a new environment may still involve limited and lightweight deployment-time calibration.
> ## W2,Q2: Edge Devices
> We thank the reviewer for this insightful comment. The $I^2O$ mainly exploits an inherent property of VLA inference, namely that the VLM backbone is more compute-intensive than the Action Expert, which leads to FLOPs bubbles that can be used to schedule the pruning in parallel. Thus, EcoVLA is general at the framework level, while hardware optimizations such as triton kernel, are CUDA-oriented. That said, transferring the EcoVLA framework to other platforms does not require redesigning the method, but mainly adapting the hardware-efficient implementation to the target runtime. To directly address this concern, we further evaluated the EcoVLA on Orin(Jetson AGX Orin Developer Kit with 64 GB RAM) and RTX 4090, and find it effective across hardware platforms.
>
> |Hardware|Sparsity|Latency(ms)|Speedup|
> |-|-|-|-|
> |Orin|0%|702.26|1.00X|
> |Orin|25%|548.67|1.28X|
> |Orin|40%|491.23|1.43X|
> |RTX 4090|0%|71.99|1.00X|
> |RTX 4090|25%|60.27|1.19X|
> |RTX 4090|40%|54.29|1.33X|
> |RTX 3090|0%|143.56|1.00X|
> |RTX 3090|25%|113.98|1.26X|
> |RTX 3090|40%|101.58|1.41X|
>
> In summary, EcoVLA can transfer across different devices, but the exact practical gains may vary depending on the target hardware and runtime characteristics.
> ## Q1: Transferability to Unified MLLMs
> We thank the reviewer for this important question. We believe EcoVLA is transferable in principle to unified multimodal models without a standalone vision encoder. While emerging unified models (e.g., Fuyu, Chameleon[1,2]) eliminate the heavy ViT, they still inherently rely on a projection layer to convert raw image patches into a sequence of tokens for the unified Transformer backbone. Because our EAP method only requires a sequence of visual tokens to compute the visual similarity for estimating environmental dynamics, we can seamlessly substitute the traditional ViT features with the image token embeddings extracted immediately after a projection layer. Therefore, EcoVLA is structurally agnostic and fully applicable to these emerging unified MLLMs.
> ### reference
> [1] Fuyu-8B: A Multimodal Architecture for AI Agents
>
> [2] Chameleon: Mixed-Modal Early-Fusion Foundation Models
> ## Q3: VRAM Usage
> We thank the reviewer for this insightful question. $I^2O$ does not introduce a large increase in peak VRAM, since it neither duplicates model weights nor keeps an extra dense model copy. Its extra memory overhead mainly comes from activation norms and precomputed weight norms. In addition, we explicitly optimize this overhead in Sec. 4.3.2. As shown by the experimental results, EcoVLA achieves substantial inference acceleration (1.41X/1.60X) with relatively limited memory overhead (around 1400/1800M), representing an effective trade-off.
> |Model|Peak VRAM(Baseline)|Peak VRAM(EcoVLA)|VRAM Increase|Speedup|
> |-|-|-|-|-|
> |OpenVLA-OFT|16564 MB|18017 MB|+1453 MB|1.41X|
> |CogACT|16720 MB|18520 MB|+1800 MB|1.6X|

---

> > ### Author Rebuttal · Reviewer_PWM5 · 2026-04-01
> >
> > The authors have addressed my primary concerns through the rebuttal. I remain positive about the paper and will maintain my current score.

---

> > > ### Author Response · Authors · 2026-04-02
> > >
> > > Thank you for your positive feedback and for taking the time to carefully read our rebuttal. We are glad that our response has addressed your primary concerns, and we sincerely appreciate your support.

---

### Official Review · Reviewer_45jg · 2026-03-11

**Soundness:** 3
**Presentation:** 3
**Significance:** 2
**Originality:** 2
**Overall Recommendation:** 4
**Confidence:** 4

**Summary:**

This paper proposes EcoVLA, a training-free, plug-and-play pruning framework that dynamically adapts sparsity patterns in response to environmental changes observed during task execution. An important concept studied by this study is environment-aware adaptive pruning, which leverages visual observations and temporal consistency across frames to identify redundant channels while maintaining stable control behavior. In addition, the framework introduces Interleaved Inference Orchestration (I2O), which schedules pruning computations within unused FLOPs during inference to minimize additional latency. Experiments on multiple VLA models, simulation benchmarks, and real-robot tasks demonstrate that EcoVLA significantly reduces inference latency while maintaining comparable task success rates, and can further improve efficiency when combined with existing acceleration methods.

**Compliance With Llm Reviewing Policy:**

Affirmed.

**Final Justification:**

I thank the authors for their thorough responses. My concerns have been fully addressed, and I am satisfied with the clarifications provided. I will accordingly raise our score to reflect this.

**Key Questions For Authors:**

- Could you explain the advantages over baselines dealing with faster inference with asynchronous inferences [1,2]?
- How is the calibration dataset constructed, and how sensitive is EcoVLA to the size/distribution of that calibration data and to benchmark-specific tuning of parameters such as the sensitivity threshold and temporal inertia?
- Can the authors provide direct comparisons to dynamic VLA efficiency baselines discussed in the paper, or explain precisely why such comparisons are infeasible? A strong apples-to-apples comparison would materially increase my confidence in the significance of the empirical gains and in the paper’s state-of-the-art framing.
- Can the authors report ablations against simpler update policies, such as fixed-interval updating, no temporal feature fusion, or always recomputing the sparsity pattern, and also provide statistics on how often the sparsity pattern is actually refreshed? This would help clarify whether the gains come specifically from the proposed temporal-consistency mechanism.
- How much of the observed speedup comes from the adaptive pruning algorithm itself versus the hardware-specific implementation choices (custom Triton kernels, memory coalescing, kernel fusion, stream orchestration), and how well does the approach transfer to other hardware? A positive answer here would substantially increase my view of the paper’s practical significance.


**References**\
[1] VLASH: Real-Time VLAs via Future-State-Aware Asynchronous Inference, ArXiv 2025.\
[2] Real-Time Execution of Action Chunking Flow Policies, ArXiv 2025.

**Limitations:**

No, this paper did not fully deal with its limitations.

**Strengths And Weaknesses:**

### Strengths
- The overall narrative is easy to follow, and the separation between the algorithmic component (EAP), the systems component (I2O), and the low-level implementation optimizations is helpful. The paper is also generally well structured.
- The paper is well motivated, and the empirical scope is broader than a typical single-model efficiency paper, including real-world robot experiments
- The individual ingredients—structured pruning, temporal smoothing of features, trigger-based adaptation, and overlapping auxiliary computation with inference—are not entirely new in isolation. However, the combination is novel in the VLA setting, and the authors articulate a clear reason for its usefulness here.

### Weaknesses
- The paper should more explicitly separate gains from the adaptive pruning idea itself from gains due to hardware-specific engineering, such as Triton kernels, memory layout changes, and fused kernels.
- Although the paper positions itself against both static and dynamic VLA pruning methods, the main quantitative comparison is relatively narrow: the tables mostly compare against Wanda and combinations with FastV/VLA-Cache, so the state-of-the-art framing feels only partially validated.
- The paper reports point estimates without uncertainty intervals, and the real-robot study uses 20 trials per task, which makes it difficult to judge whether small reported differences are reliable.
- The paper would be stronger if it more directly demonstrated that the proposed trigger/reuse mechanism outperforms simpler alternatives such as fixed-interval updates or variants without temporal fusion.

---

> ### Author Rebuttal · Authors · 2026-03-31
>
> # Response to Reviewer 45jg
> We thank the reviewer for the insightful feedback. Unless otherwise noted, all rebuttal experiments are conducted on OpenVLA-OFT in LIBERO-Goal.
> ## W1,Q5: EAP Gain and Transferability
> We agree that the gains from EAP and hardware optimizations should be separated. However, **in VLA inference, FLOPs reduction and wall-clock latency improvement are not equivalent.** EAP reduces FLOPs at the algorithmic level, but this does not automatically translate into wall-clock speedup. To our knowledge, it is the first work on adaptive pruning for VLA, and $I^2O$ together with the hardware-level optimizations play a key supporting role in making it practical. Following the reviewer's suggestion, we measure the gain brought by EAP alone on an RTX 4090 as table below. Moreover, **Fig.4 in the paper decomposes the end-to-end latency gains from $I^2O$ with hardware-level optimizations.**
>
> |Hardware|Sparsity|Latency(ms)|Speedup|
> |-|-|-|-|
> |Orin|0%|702.26|1.00X|
> |Orin|25%|548.67|1.28X|
> |Orin|40%|491.23|1.43X|
> |RTX 4090|0%|71.99|1.00X|
> |RTX 4090(EAP only)|25%|66.28|1.09X|
> |RTX 4090|25%|60.27|1.19X|
> |RTX 4090|40%|54.29|1.33X|
>
> As for transferability, EAP is algorithm-level and hardware-agnostic. $I^2O$ relies on a general property of VLA inference, namely the compute imbalance between the VLM and the Action Expert. Thus, EcoVLA is general at the framework level, while hardware optimizations such as triton kernel, are CUDA-oriented. **Porting EcoVLA to other platforms mainly requires implementation adaptation rather than method redesign**. We evaluate EcoVLA on Orin(Jetson AGX Orin Developer Kit with 64 GB RAM) and RTX 4090, and find it effective across hardware platforms.
>
> ## W2,Q3: Baseline
> We use Wanda as the main model pruning baseline because it's the SOTA training-free pruning method for VLA. **Existing dynamic VLA pruning methods require additional training or auxiliary learned modules, so they are not directly comparable in a true apples-to-apples setting.** We also evaluated other training-free methods designed for LLM/VLM, but their transfer to VLA tasks was less consistent and resulted in lower success rates, which makes them less suitable.
> |Method|Type|Sparsity|SR|
> |-|-|-|-|
> |FLAP|Model pruning|25%|76.4|
> |DivPrune|Token pruning|25%|47.2|
> ## W3: Uncertainty and Real-robot Experiments
> Following the suggestion, we conduct three repeated runs on each dataset and report the corresponding uncertainty statistics.
>
> ||spatial|object|goal|long|
> |-|-|-|-|-|
> |SR|$98.13\pm0.73$|$98.80\pm0.20$|$94.60\pm0.40$|$96.53\pm0.27$|
>
> For the real-robot study, we ran 20 trials per task due to resource constraints. We further conduct real-robot experiments on Franka Research 3 with XHand([setup figure](https://anonymous.4open.science/r/anon-repo-9663/franka_xhand.pdf)).
>
> |Method|pickup reagent bottle|stack white block on green|put block in the basket|Latency|
> |-|-|-|-|-|
> |$\pi_{0.5}$|27/50|23/50|25/50|81.81|
> |Ours(25%)|26/50|22/50|25/50|62.53|
> ## W4,Q4: Update Policy
> We first measured EAP's actual sparsity pattern update frequency and found it updates once every 6.94 steps on average. We then compare EAP against fixed-interval update strategies. Third, we evaluate a variant without temporal fusion.
> |Method|SR|
> |-|-|
> |EAP(6.94)|94.6|
> |always recompute|92.0|
> |fixed-interval(every 3 step)|92.2|
> |5 step|93.2|
> |7 step|91.2|
> |w/o temporal fusion|90.6|
> ## Q1: Asynchronous Inference
> We agree that asynchronous inference is relevant to real-time VLA deployment, but would like to clarify that such methods and ours belong to two different directions. Methods such as VLASH mainly address the prediction-execution misalignment introduced by asynchronous inference. In contrast, EcoVLA reduces the per-inference cost through adaptive pruning. **Thus, the two are complementary: asynchronous inference changes how inference is scheduled relative to execution, while EcoVLA reduces the cost of each inference step.** In principle, **EcoVLA can also be combined with asynchronous inference methods**. Therefore, we view the two as complementary rather than strictly competing. We'll discuss these works in the revision.
> ## Q2: Calib Data and Parameter Sensitivity
> Our calibration data is built by randomly sampling 1,024 examples from the training data. We further evaluate different calibration sizes, random seeds, and find EcoVLA remains stable across these settings. Fig.6 and App.C analyze parameter sensitivity, showing that EcoVLA is not overly sensitive to precise hyperparameter tuning, but instead remains stable and effective across a reasonable range. Since EcoVLA is training-free and these parameters have intuitive meanings, the adjustment is low-cost. To support this, we additionally conduct experiments on OpenVLA-OFT as summarized in [figure](https://anonymous.4open.science/r/anon-repo-9663/temporal_sensitivity.pdf).
> |Data Size|SR|
> |-|-|
> |256|93.2|
> |512|94.2|
> |1024|94.6|
> |2048|94.4|
> |Construction Setting||
> |seed=43|94.6|
> |seed=44|94.4|

---

> > ### Author Rebuttal · Reviewer_45jg · 2026-04-03
> >
> > I thank the authors for their thorough responses. My concerns have been fully addressed, and I am satisfied with the clarifications provided. I will accordingly raise the score to reflect this.

---

> > > ### Author Response · Authors · 2026-04-03
> > >
> > > We sincerely appreciate your valuable feedback and your careful reading of our rebuttal. We are glad that our clarifications have fully addressed your concerns, and we are truly grateful for your support and for raising the score.

---

### Official Review · Reviewer_qPTj · 2026-03-13

**Soundness:** 3
**Presentation:** 2
**Significance:** 3
**Originality:** 3
**Overall Recommendation:** 4
**Confidence:** 4

**Summary:**

This paper introduces EcoVLA, a training-free and plug-and-play adaptive pruning framework designed to accelerate Vision-Language-Action (VLA) models for real-time robotic manipulation. The proposed method is designed to address the limitations of static pruning, which fails to adapt to dynamic environments, and fixed-interval dynamic pruning, which introduces high overhead. Overall the experimental results reported are strong, but they also exposure clear limitation on the method and induce further concerns. The paper struck as a piece of work that has clear advantages and disadvantages.

**Compliance With Llm Reviewing Policy:**

Affirmed.

**Final Justification:**

The authors provided further explanations and experimental results, which fully addressed my concerns. I've raised my score.

**Key Questions For Authors:**

- Can you provide more clarity on the issues I raised with Weakness#2 and #3?
- What's the implications on memory complexity I^2O? Does the Shared Memory add significant memory requirement?
- Since channel-based model pruning is not new, would you agree that the biggest contribution of this paper is the novel I^2O pipeline that fully leverages the computational capacity and makes channel sparcification feasible for VLA models?

**Limitations:**

- Suggest to discuss possibility/feasibility of adopting finetune-free or training-free pruning methods, which are plenty in the ViT/Transfromer pruning community.

**Strengths And Weaknesses:**

Strengths:
- Model pruning for VLA models is a timely and important research area given the increasing research effort going into embodied AI and it is natural to devise and apply model pruning techniques onto VLA models.
- Strong experimental results, especially on pi_0.5 which is already an efficiency-optimized model.
- The proposed pipeline being a plug n' play, generic solution to various VLA models.

Weaknesses:
- The biggest concern is the degradation to edge cases and boundary condition, where the pruned models seem to struggle with. Although on the public benchmarks the SR drops are reasonable, the results indicate a decreased generalizability of the models. This might pose challenges in real-life applications.
- In real-life experiments, the delta in the latency is much more marginal than in simulations. Any insights?
- The presentation and interpretation of the results can be improved, especially around the meaning of the speedup ratios and latency. For example, in the experiment section, Table 4 is not not referenced and interpreted. For Table 5, it lacks clarity on what it means for a higher latency with I^2O than normal VLA inference. Figure 4 was a bit confusing with a stacked-bar plot as the elements on a bar represent different meanings, i.e. actual run time vs. time savings. Maybe think about a better way to visualize the component-wise time savings to avoid confusion. Also for this figure, does the "original latency" mean a pruning pipeline without parallelization (I^2O) or the original, single-model VLA inference?
- From the Supplementary Materials,  it seems that the pipeline requires significant finetuning effort. However there is a  collection of finetune-free or training-free, channel-based pruning methods from the ViT/Transfromer pruning community, which has not been discussed or considered.

---

> ### Author Rebuttal · Authors · 2026-03-31
>
> # Response to Reviewer qPTj
> Thanks for the positive comments and constructive suggestions.
> ## W4: Training-Free
> We apologize for the ambiguity. To clarify, **EcoVLA itself requires neither training nor finetuning**. It is an inference-time acceleration method. The finetuning mentioned in the Supplementary Material is **not part of EcoVLA**; it is only used to adapt the base model $\pi_{0.5}$ for deployment on the Kinova Gen3 platform, since the released pretrained checkpoint cannot be directly executed on that setup.
>
> We appreciate the reviewer’s suggestion regarding related methods for ViT/Transformer. This work targets training-free efficiency in the VLA, and we will clarify this distinction and broaden the related-work discussion accordingly.
> ## W1: Boundary Conditions
> As discussed in Appendix B, EcoVLA shows some degradation only under extreme boundary conditions, rather than a general loss of generalization across tasks. **Moreover, the practical importance of such extreme boundary conditions depends on the deployment setting: in many structured industrial scenarios, extreme boundary conditions much less common, while low latency is often especially valuable for throughput and efficiency.** More broadly, pruning should be viewed as a trade-off between robustness in challenging boundary conditions and inference efficiency, rather than optimizing only one side. We also agree that improving pruning robustness under boundary conditions is an important direction for future work.
> ## W2 and Q1: Real-robot Experiments
> The original experiments were conducted on distinct physical machines, which may differ in CPU architecture, memory bandwidth, and PCIe configuration. To reduce this confound, we conduct additional real-robot experiments on the same machine using the Franka Research 3 with XHand platform, achieving a 1.31X speedup. The detailed setup is provided at [setup figure](https://anonymous.4open.science/r/anon-repo-9663/franka_xhand.pdf)
> |Method|pickup reagent bottle|stack white block on green|put block in the basket|Latency|
> |-|-|-|-|-|
> |$\pi_{0.5}$|27/50|23/50|25/50|81.81|
> |Ours(25%)|26/50|22/50|25/50|62.53|
> ## W3 and Q1: Presentation Issues
> Regarding Table 4, it's a typo, and the statement in the *Results on Real Robot* section should refer to Table 4 rather than Table 3. We will correct this in the revision. Regarding Table 5, its purpose is to show the overhead $\delta$ incurred by sparsity pattern computation during dense inference. Due to the lightweight pruning module and its execution within FLOPs bubbles, $\delta$ is only 4.5 ms relative to the overall inference time. We have analyzed the source and magnitude of $\delta$ theoretically in Section 4.2.2, and we will reference this analysis near Table 5 in the revision. Regarding Figure 4, it's intended to provide an end-to-end latency breakdown of the gains from $I^2O$ with hardware-level optimizations in dense and sparse inference. We agree that the current visualization is not intuitive, and we have redrawn it as a clearer table. Finally, 'original latency' refers to the EcoVLA without any hardware-level optimizations, not the base VLA model inference.
> |Setting|Latency breakdown|Latency(ms)|
> |-|-|-|
> |Dense inference| Original latency|215.24|
> ||+Parallel paradigm|-36.04|
> ||+Allocation-free caching|-10.00|
> ||+Batched metric computation|-21.14|
> ||Final latency|148.06|
> |Sparse inference|Original latency|156.12|
> ||+Sparse linear transformation kernel|-32.63|
> ||+Memory coalescing|-13.49|
> ||+Kernel fusion|-1.76|
> ||Final latency|108.24|
> ## Q2: VRAM Usage
> $I^2O$ does not significantly increase memory, since it neither duplicates model weights nor keeps an extra dense model copy. Its memory overhead mainly comes from activation norms and precomputed weight norms. Moreover, the overhead is explicitly optimized in Section 4.3.2. As shown by the experimental results, EcoVLA achieves substantial inference acceleration (1.41X/1.60X) with relatively limited memory overhead (around 1400/1800M), representing an effective trade-off.
> |Model|Peak VRAM(Baseline)|Peak VRAM(EcoVLA)|VRAM Increase|Speedup|
> |-|-|-|-|-|
> |OpenVLA-OFT|16564 MB|18017 MB|+1453 MB|1.41X|
> |CogACT|16720 MB|18520 MB|+1800 MB|1.6X|
> ## Q3: EcoVLA Novelty
> We thank the reviewer for recognizing the novelty of $I^2O$. We agree that model pruning is not new. However, we would like to clarify that our work is not a simple combination of 'a pruning method' and 'a system module'. Instead, **EcoVLA is an algorithm-system co-design tailored to the runtime characteristics of VLA models.** EAP leverages the temporal consistency of VLA execution to perform lightweight yet effective adaptive pruning, while $I^2O$ exploits the resource usage characteristics of VLA inference to translate the FLOPs reduction into practical latency reduction. To our knowledge, it's the first framework to support real-time adaptive pruning for VLA through joint algorithm-system co-design.

---

> > ### Author Rebuttal · Reviewer_qPTj · 2026-04-01
> >
> > My questions are answered and I have no more questions.

---

> > > ### Author Response · Authors · 2026-04-02
> > >
> > > Thank you very much for your positive update. We are very glad that our rebuttal has addressed your concerns and that you have no further questions.
> > >
> > > Given your current assessment, if you find it appropriate, we would greatly appreciate it if the score could be updated accordingly.
> > >
> > > Thank you again for your careful reading and valuable feedback.

---

### Decision · Program_Chairs · 2026-04-30

**Decision:**

Accept (spotlight)

**Comment:**

This paper introduces a training-free adaptive pruning framework for VLAs. Reviewers appreciated the problem statement, plug-and-play nature of the method, and strong experimental results. They also identified several concerns, including degradation on edge cases, latency delta in real-world experiments, narrowness of the evaluation, and overall presentation of the paper. After the extensive rebuttal, all reviewers recommended acceptance of this paper and mentioned that all of their concerns were fully addressed. As a result, I recommend acceptance and recommend that the authors include rebuttal materials in the camera-ready version.